# REFERENCE-AWARE LANGUAGE MODELS

**Zichao Yang**[1*]**, Phil Blunsom**[2,3]**, Chris Dyer**[1,2]**, and Wang Ling**[2]
[1]Carnegie Mellon University, [2]DeepMind, and [3]University of Oxford
`zichaoy@cs.cmu.edu`, `{pblunsom,cdyer,lingwang}@google.com`

## ABSTRACT

We propose a general class of language models that treat reference as an explicit stochastic latent variable. This architecture allows models to create mentions of entities and their attributes by accessing external databases (required by, e.g., dialogue generation and recipe generation) and internal state (required by, e.g. language models which are aware of coreference). This facilitates the incorporation of information that can be accessed in predictable locations in databases or discourse context, even when the targets of the reference may be rare words. Experiments on three tasks show our model variants outperform models based on deterministic attention.

## 1 INTRODUCTION

Referring expressions (REs) in natural language are noun phrases (proper nouns, common nouns, and pronouns) that identify objects, entities, and events in an environment. REs occur frequently and they play a key role in communicating information efficiently. While REs are common, previous works neglect to model REs explicitly, either treating REs as ordinary words in the model or replacing them with special tokens. Here we propose a language modeling framework that explicitly incorporates reference decisions.

In Figure 1 we list examples of REs in the context of the three tasks that we consider in this work. Firstly, reference to a database is crucial in many applications. One example is in task oriented dialogue where access to a database is necessary to answer a user's query (Young et al., 2013; Li et al., 2016; Vinyals & Le, 2015; Wen et al., 2015; Sordoni et al., 2015; Serban et al., 2016; Bordes & Weston, 2016; Williams & Zweig, 2016; Shang et al., 2015; Wen et al., 2016). Here we consider the domain of restaurant recommendation where a system refers to restaurants (name) and their attributes (address, phone number etc) in its responses. When the system says "`the nirala` is a nice restaurant", it refers to the restaurant name `the nirala` from the database. Secondly, many models need to refer to a list of items (Kiddon et al., 2016; Wen et al., 2015). In the task of recipe generation from a list of ingredients (Kiddon et al., 2016), the generation of the recipe will frequently reference these items. As shown in Figure 1, in the recipe "Blend `soy milk` and...", `soy milk` refers to the ingredient summaries. Finally, we address references within a document (Mikolov et al., 2010; Ji et al., 2015; Wang & Cho, 2015), as the generation of words will ofter refer to previously generated words. For instance the same entity will often be referred to throughout a document. In Figure 1, the entity `you` refers to `I` in a previous utterance.

In this work we develop a language model that has a specific module for generating REs. A series of latent decisions (should I generate a RE? If yes, which entity in the context should I refer to? How should the RE be rendered?) augment a traditional recurrent neural network language model and the two components are combined as a mixture model. Selecting an entity in context is similar to familiar models of attention (Bahdanau et al., 2014), but rather than being a deterministic function that reweights representations of elements in the context, it is treated as a distribution over contextual elements which are stochastically selected and then copied or, if the task warrants it, transformed (e.g., a pronoun rather than a proper name is produced as output). Two variants are possible for updating the RNN state: one that only looks at the generated output form; and a second that looks at values of the latent variables. The former admits trivial unsupervised learning, latent decisions are conditionally independent of each other given observed context, whereas the latter enables more

---

*Work completed at DeepMind.

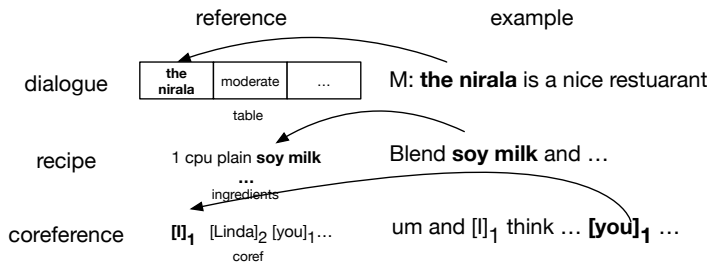

Figure 1: Reference-aware language models.

expressive models that can extract information from the entity that is being referred to. In each of the three tasks, we demonstrate our reference aware model's efficacy in evaluations against models that do not explicitly include a reference operation.

Our contributions are as follows:

- We propose a general framework to model reference in language and instantiate it in the context of dialogue modeling, recipe generation and coreference based language models.
- We build three data sets to test our models. There lack existing data sets that satisfy our need, so we build these data sets ourselves. These data sets are either built on top existing data set (we constructed the table for DSTC2 data set for dialogue evaluation), crawled from websites (we crawled all recipes in www.allrecipes.com) or annotated with NLP tools (we annotate the coreference with Gigaword corpus for our evaluation).
- We perform comprehensive evaluation of our models on the three data sets and verify our models perform better than strong baselines.

## 2  REFERENCE-AWARE LANGUAGE MODELS

Here we propose a general framework for reference-aware language models.

We denote each document as a series of tokens $x_1, \ldots, x_L$, where $L$ is the number of tokens in the document. Our goal is to maximize the probabilities $p(x_i \mid c_i)$, for each word in the document based on its previous context $c_i = x_1, \ldots, x_{i-1}$. In contrast to traditional neural language models, we introduce a variable at each position $z_i$, which controls the decision on which source $x_i$ is generated from. The token conditional probably is then obtained by:

$$p(x_i \mid c_i) = p(x_i \mid z_i, c_i)p(z_i \mid c_i). \tag{1}$$

In dialogue modeling and recipe generation, $z_i$ will simply taken on values in $\{0, 1\}$. Where $z_i = 1$ denotes that $x_i$ is generated as a reference, either to a database entry or an item in a list. However, $z_i$ can also be defined as a distribution over previous entities, allowing the model to predict $x_i$ conditioned on its a previous mention word. This will be the focus of the coreference language model. When $z_i$ is not observed (which it generally will not be), we will train our model to maximize the marginal probability in Eq. 1 directly.

### 2.1  DIALOGUE MODEL WITH DATABASE SUPPORT

We first apply our model on task-oriented dialogue systems in the domain of restaurant recommendations, and work on the data set from the second Dialogue State Tracking Challenge (DSTC2) (Henderson et al., 2014). Table. 1 is one example dialogue from this dataset.

We can observe from this example, users get recommendations of restaurants based on queries that specify the area, price and food type of the restaurant. We can support the system's decisions by incorporating a mechanism that allows the model to query the database allowing the model to find restaurants that satisfy the users queries. Thus, we crawled TripAdvisor for restaurants in the

| | | | | | | | |
|---|---|---|---|---|---|---|---|
| M: | Hello , welcome to the Cambridge restaurant system? You can ask for restaurants by `area`, `price range` or `food type`. How may I help you? |
| U: | i want a `moderately priced` restaurant in the `north` town |
| M: | `the nirala` is a nice restaurant in the north of town in the moderate price range |
| U: | what is the `address` |
| M: | Sure , `the nirala` is on `7 Milton Road Chesterton` |

Table 1: Example dialogue, M stands for Machine and U stands for User

| NAME | PRICE RANGE | FOOD | AREA | ADDRESS | POST CODE | PHONE |
|---|---|---|---|---|---|---|
| ali baba | moderate | lebanese | centre | 59 Hills Road City Centre | CB 2, 1 NT | 01462 432565 |
| the nirala | moderate | indian | north | 7 Milton Road Chesterton | CB 4, 1 UY | 01223 360966 |

Table 2: Fragment of database for dialogue system.

Cambridge area, where the dialog dataset was collected. Then, we remove restaurants that do not appear in the data set and create a database with 109 entries with restaurants and their attributes (e.g. food type). A sample of our database is shown in Table. 2. We can observe that each restaurant contains 6 attributes that are generally referred in the dialogue dataset. As such, if the user requests a restaurant that serves "indian" food, we wish to train a model that can search for entries whose "food" column contains "indian". Now, we describe how we deploy a model that fulfills these requirements.

### 2.1.1 DIALOGUE MODEL

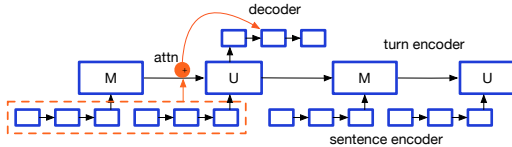

Figure 2: Hierarchical RNN Seq2Seq model

We build a model based on the hierarchical RNN model described in (Serban et al., 2016), as in dialogues, the generation of the response is not only dependent on the previous sentence, but on all sentences leading to the response. We assume that a dialogue is alternated between a machine and a user. An illustration of the model is shown in Figure 2.

Consider a dialogue with $T$ turns, and the utterance from a user is denoted as $X = \{x_i\}_{i=1}^T$, where $i$ is the $i$-th utterance, whereas the utterance from a machine is denoted as $Y = \{y_i\}_{i=1}^T$, where $i$ is the $i$-th utterance. We define $x_i = \{x_{ij}\}_{j=1}^{|x_i|}$, $y_i = \{y_{iv}\}_{v=1}^{|y_i|}$, where $x_{ij}$ denotes the $j$-th token in the $i$-th utterance from the user, whereas $y_{iv}$ denotes the $v$-th token in the $i$-th utterance from the machine. Finally, $|x_i|$ and $|y_i|$ denote the number of tokens in the user and machine utterances, respectively. The dialogue sequence starts with machine utterance $\{y_1, x_1, y_2, x_2, \ldots, y_T, x_T\}$. We would like to model the utterances from the machine

$$p(y_1, y_2, \ldots, y_T | x_1, x_2, \ldots, x_T) = \prod_i p(y_i | y_{<i}, x_{<i}) = \prod_{i,v} p(y_{i,v} | y_{i,<v}, y_{<i}, x_{<i}),$$

where $y_{<i}$ denotes all the utterances before $i$ and $y_{i,<v}$ denotes the first $v - 1$ tokens in the $i$-th utterance of the machine. A neural model is employed to predict $p(y_{i,v} | y_{i,<v}, y_{<i}, x_{<i})$, which operates as follows:

**Sentence Encoder**: We first encode previous utterances $y_{<i}$ and $x_{<i}$ into continuous space by generating employing a LSTM encoder. Thus, for a given utterance $x_i$, and start with the initial LSTM state $h_{i,0}^x$ and apply the recursion $h_{i,j}^x = \text{LSTM}_\text{E}(W_E x_{i,j}, h_{i,j-1}^x)$, where $W_E x_{i,j}$ denotes a word

embedding lookup for the token $x_{i,j}$, and $\text{LSTM}_\text{E}$ denotes the LSTM transition function described in Hochreiter & Schmidhuber (1997). The representation of the user utterance is represented by the final LSTM state $h_i^x = h_{i,|x_i|}^x$. The same process is applied to obtain the machine utterance representation $h_i^y = h_{i,|y_i|}^y$.

**Turn Encoder**: Then, combine all the representations of all the utterances with a second LSTM, which encodes the sequence $\{h_1^y, h_1^x, ..., h_i^y, h_i^x\}$ into a continuous vector. Once again, we start with an initial state $u_0$ and feed each of the utterance representation to obtain the following LSTM state, until the final state is obtained. For simplicity, we shall refer to this as $u_i$, which can be seen as the hierarchical encoding of the previous $i$ utterances.

**Seq2Seq Decoder**: As for decoding, in order to generate each utterance $y_i$, we can feed $u_{i-1}$ into the decoder LSTM as the initial state $s_{i,0} = u_{i-1}$ and decode each token in $y_i$. Thus, we can express the decoder as:

$$s_{i,v}^y = \text{LSTM}_\text{D}(W_E y_{i,v-1}, s_{i,v-1}),$$
$$p_{i,v}^y = \text{softmax}(W s_{i,v}^y),$$

where the desired probability $p(y_{i,v}|y_{i,<v}, y_{<i}, x_{<i})$ is expressed by $p_{i,v}^y$.

**Attention based decoder**: We can also incorporate the attention mechanism in our hierarchical model. An attention model builds a representation $d$ by averaging over a set of vectors $p$. We define the attention function as $a = \text{ATTN}(p, q)$, where $a$ is a probability distribution over the set of vectors $p$, conditioned on any input representation $q$. A full description of this operation is described in (Bahdanau et al., 2014). Thus, for each generated token $y_{i,v}$, we compute the attentions $a_{i,v}$, conditioned on the current decoder state $s_{i,v}^y$, obtaining the attentions over input tokens from previous turn $(i-1)$. We denote the vector of all tokens in previous turn as $h_{i-1}^{x,y} = [\{h_{i-1,j}^x\}_{j=1}^{|x_{i-1}|}, \{h_{i-1,v}^y\}_{v=1}^{|y_{i-1}|}]$. Let $K = |h_{i-1}^{x,y}|$ be the number of tokens in previous turn. Thus, we obtain the attention probabilities over all previous tokens $a_{i,v}$ as $\text{ATTN}(s_{i,v}^y, h_{i-1}^{x,y})$. Then, the weighted sum is computed over these probabilities $d_{i,v} = \sum_{k \in K} a_{i,v,k} h_{i-1,k}^{x,y}$, where $a_{i,v,k}$ is the probability of aligning to the $k$-th token from previous turn. The resulting vector $d_{i,v}$ is used to obtain the probability of the following word $p_{i,v}^y$. Thus, we express the decoder as:

$$s_{i,v}^y = \text{LSTM}_\text{D}([W_\text{E} y_{i,v-1}, d_{i,v-1}], s_{i,v-1}),$$
$$a_{i,v} = \text{ATTN}(h_{i-1}^{x,y}, s_{i,v}^y),$$
$$d_{i,v} = \sum_{k \in K} a_{i,v,k} h_{i-1,k}^{x,y},$$
$$p_{i,v}^y = \text{softmax}(W[s_{i,v}^y, d_{i,v}]).$$

### 2.1.2 INCORPORATING TABLE ATTENTION

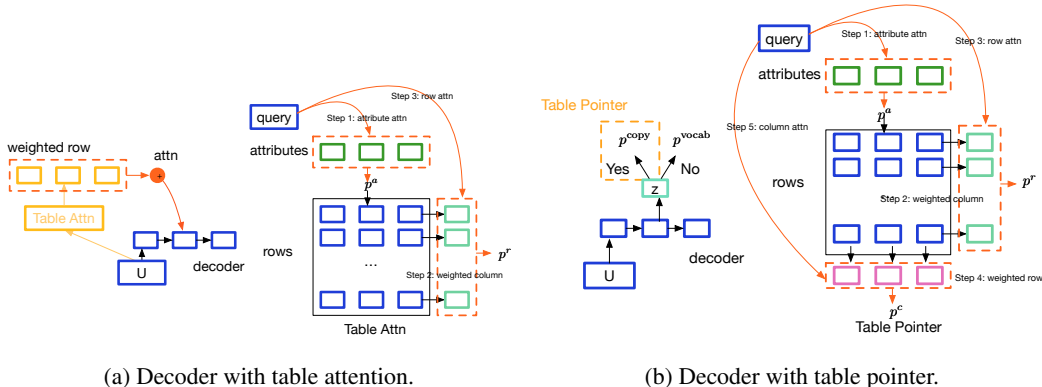

(a) Decoder with table attention.  (b) Decoder with table pointer.

Figure 3: Table based decoder.

We now extend the attention model in order to allow the attention to be computed over a table, allowing the model to condition the generation on a database.

We denote a table with $R$ rows and $C$ columns as $\{f_{r,c}\}, r \in [1, R], c \in [1, C]$, where $f_{r,c}$ is the cell in row $r$ and column $c$. The attribute of each column is denoted as $s_c$, where $c$ is the $c$-th attribute. $f_{r,c}$ and $s_c$ are one-hot vector.

**Table Encoding**: To encode the table, we build an attribute vector $g_c$ for each column. For each cell $f_{r,c}$ of the table, we concatenate it with the corresponding attribute $g_c$ and then feed it through a one-layer MLP as follows: $g_c = W_E s_c$ and then $e_{r,c} = \tanh(W[W_E f_{r,c}, g_c])$.

**Table Attention**: The diagram for table attention is shown in Figure 3a. The attention over cells in the table is conditioned on a given vector $q$, similarly to the attention model for sequences ATTN$(p, q)$. However, rather than a sequence $p$, we now operate over a table $f$. Our attention model computes a attribute attention followed by row attention of the table. We first use the attention mechanism on the attributes to find out which attribute the user asks about. Suppose a user says cheap, then we should focus on the price attribute. After we get the attention probability $p^a = \text{ATTN}(\{g_c\}, q)$, over the attribute, we calculate the weighted representation for each row $e_r = \sum_c p_c^a e_{rc}$ conditioned on $p^a$. Then $e_r$ has the price information of each row. We further use attention mechanism on $e_r$ and get the probability $p^r = \text{ATTN}(\{e_r\}, q)$ over the rows. Then restaurants with cheap price will be picked. Then, using the probabilities $p^r$, we compute the weighted average over the all rows $e_c = \sum_r p_r^r e_{r,c}$, which is used in the decoder. The detailed process is:

$$p_a = \text{ATTN}(\{g_c\}, q), \tag{2}$$

$$e_r = \sum_c p_c^a e_{rc} \quad \forall r, \tag{3}$$

$$p_r = \text{ATTN}(\{e_r\}, q), \tag{4}$$

$$e_c = \sum_r p_r^r e_{r,c} \quad \forall c. \tag{5}$$

This is embedded in the decoder by replacing the conditioned state $q$ as the current decoder state $s_{i,0}^y$ and then at each step, conditioning the prediction of $y_{i,v}$ on $\{e_c\}$ by using attention mechanism at each step. The detailed diagram of table attention is shown in Figure 3a.

### 2.1.3 Incorporating Table Pointer Networks

We now describe the mechanism used to refer to specific database entries during decoding. At each timestep, the model needs to decide whether to generate the next token from an entry of the database or from the word softmax. This is performed as follows.

**Pointer Switch**: We use $z_{i,v} \in [0, 1]$ to denote the decision of whether to copy one cell from the table. We compute this probability as follows:

$$p(z_{i,v}|s_{i,v}) = \text{sigmoid}(W[s_{i,v}, d_{i,v}]).$$

Thus, if $z_{i,v} = 1$, the next token $y_{i,v}$ will be generated from the database, whereas if $z_{i,v} = 0$, then the following token is generated from a softmax. We shall now describe how we generate tokens from the database.

**Table Pointer**: If $z_{i,v} = 1$, the token is generated from the table. The detailed process of calculating the probability distribution over the table is shown in Figure 3b. This is similar to the attention mechanism, except that we perform a column attention to compute the probabilities of copying from each column after Equation. 5. More formally:

$$p^c = \text{ATTN}(\{e_c\}, q), \tag{6}$$

$$p^{\text{copy}} = p^r \otimes p^c, \tag{7}$$

where $p^c$ is a probability distribution over columns, whereas $p^r$ is a probability distribution over rows. In order to compute a matrix with the probability of copying each cell, we simply compute the outer product $p^{\text{copy}} = p^r \otimes p^c$.

**Objective:** As we treat $z_i$ as a latent variable, we wish to maximize the marginal probability of the sequence $y_i$ over all possible values of $z_i$. Thus, our objective function is defined as:

$$p(y_{i,v}|s_{i,v}) = p^{\text{vocab}}p(0|s_{i,v}) + p^{\text{copy}}p(1|s_{i,v}) = p^{\text{vocab}}(1 - p(1|s_{i,v})) + p^{\text{copy}}p(1|s_{i,v}). \tag{8}$$

The model can also be trained in a fully supervised fashion, if $z_{i,v}$ is observed. In such cases, we simply maximize the likelihood of $p(z_{i,v}|s_{i,v})$, based on the observations, rather than using the marginal probability over $z_{i,v}$.

## 2.2 RECIPE GENERATION

| ingredients | recipe |
|---|---|
| 1 cup plain `soy milk` <br> 3/4 cup packed fresh `spinach leaves` <br> 1 large `banana`, sliced | Blend `soy milk` and `spinach leaves` together in a blender until smooth. Add `banana` and pulse until thoroughly blended. |

Table 3: Ingredients and recipe for *Spinach and Banana Power Smoothie*.

Next, we consider the task of recipe generation conditioning on the ingredient lists. In this task, we must generate the recipe from a list of ingredients. Table. 3 illustrates the ingredient list and recipe for *Spinach and Banana Power Smoothie*. We can see that the ingredients `soy milk, spinach leaves, and banana` occur in the recipe.

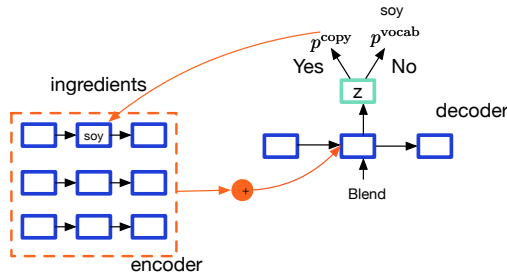

Figure 4: Recipe pointer

Let the ingredients of a recipe be $X = \{x_i\}_{i=1}^T$ and each ingredient contains $L$ tokens $x_i = \{x_{ij}\}_{j=1}^L$. The corresponding recipe is $y = \{y_v\}_{v=1}^K$. We first use a LSTM to encode each ingredient:

$$h_{i,j} = \text{LSTM}_E(W_E x_{ij}, h_{i,j-1}) \quad \forall i.$$

Then, we sum the resulting state of each ingredient to obtain the starting LSTM state of the decoder. Once again we use an attention based decoder:

$$s_v = \text{LSTM}_D(s_{v-1}, d_{v-1}, W_E y_{v-1}),$$
$$p_v^{\text{copy}} = \text{ATTN}(\{\{h_{i,j}\}_{i=1}^T\}_{j=1}^L, s_v),$$
$$d_v = \sum_{ij} p_{v,i,j} h_{i,j},$$
$$p(z_v|s_v) = \text{sigmoid}(W[s_v, d_v]),$$
$$p_v^{\text{vocab}} = \text{softmax}(W[s_v, d_v]).$$

Similar to the previous task, the decision to copy from the ingredient list or generate a new word from the softmax is performed using a switch, denoted as $p(z_v|s_v)$. We can obtain a probability distribution of copying each of the words in the ingredients by computing $p_v^{\text{copy}} = \text{ATTN}(\{\{h_{i,j}\}_{i=1}^T\}_{j=1}^L, s_v)$ in the attention mechanism. For training, we optimize the marginal likelihood function employed in the previous task.

## 2.3 COREFERENCE BASED LANGUAGE MODEL

Finally, we build a language model that uses coreference links to point to previous words. Before generating a word, we first make the decision on whether it is an entity mention. If so, we decide

which entity this mention belongs to, then we generate the word based on that entity. Denote the document as $X = \{x_i\}_{i=1}^L$, and the entities are $E = \{e_i\}_{i=1}^N$, each entity has $M_i$ mentions, $e_i = \{m_{ij}\}_{j=1}^{M_i}$, such that $\{x_{m_{ij}}\}_{j=1}^{M_i}$ refer to the same entity. We use a LSTM to model the document, the hidden state of each token is $h_i = \text{LSTM}(W_E x_i, h_{i-1})$. We use a set $h^e = \{h_0^e, h_1^e, ..., h_M^e\}$ to keep track of the entity states, where $h_j^e$ is the state of entity $j$.

um and $[\text{I}]_1$ think that is whats - Go ahead $[\text{Linda}]_2$. Well and thanks goes to $[\text{you}]_1$ and to $[\text{the media}]_3$ to help $[\text{us}]_4$...So $[\text{our}]_4$ hat is off to all of $[\text{you}]_5$...

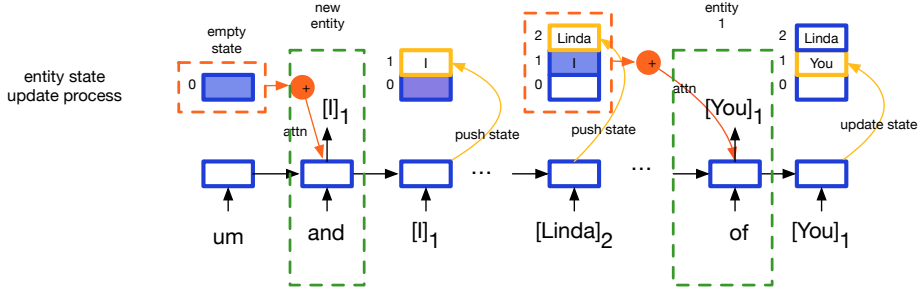

Figure 5: Coreference based language model, example taken from Wiseman et al. (2016).

**Word generation**: At each time step before generating the next word, we predict whether the word is an entity mention:

$$p^{\text{coref}}(v_i|h_{i-1}, h^e) = \text{ATTN}(h^e, h_{i-1}),$$

$$d_i = \sum_{v_i} p(v_i)h_{v_i}^e$$

$$p(z_i|h_{i-1}) = \text{sigmoid}(W[d_i, h_{i-1}]),$$

where $z_i$ denotes whether the next word is an entity and if yes $v_i$ denotes which entity the next word corefers to. If the next word is an entity mention, then $p(x_i|v_i, h_{i-1}, h^e) = \text{softmax}(W_1 \tanh(W_2[h_{v_i}^e, h_{i-1}]))$ else $p(x_i|h_{i-1}) = \text{softmax}(W_1 h_{i-1})$,

$$p(x_i|x_{<i}) = \begin{cases} p(x_i|h_{i-1})p(z_i|h_{i-1}, h^e) & \text{if} \quad z_i = 0. \\ p(x_i|v_i, h_{i-1}, h^e)p^{\text{coref}}(v_i|h_{i-1}, h^e)p(z_i|h_{i-1}, h^e) & \text{if} \quad z_i = 1. \end{cases} \quad (9)$$

**Entity state update**: We update the entity state $h^e$ at each time step. In the beginning, $h^e = \{h_0^e\}$, $h_0^e$ denotes the state of an virtual empty entity and is a learnable variable. If $z_i = 1$ and $v_i = 0$, then it indicates the next word is a new entity mention, then in the next step, we append $h_i$ to $h^e$, i.e., $h^e = \{h^e, h_i\}$, if $e_i > 0$, then we update the corresponding entity state with the new hidden state, $h^e[v_i] = h_i$. Another way to update the entity state is to use one LSTM to encode the mention states and get the new entity state. Here we use the latest entity mention state as the new entity state for simplicity. The detailed update process is shown in Figure 5.

## 3 EXPERIMENTS

## 4 DATA SETS AND PREPROCESSING

**Dialogue**: We use the DSTC2 data set. We only extracted the dialogue transcript from data set. There are about 3,200 dialogues in total. Since this is a small data set, we use 5-fold cross validation and report the average result over the 5 partitions. There may be multiple tokens in each table cell, for example in Table.2, the name, address, post code and phone number have multiple tokens, we replace them with one special token. For the name, address, post code and phone number of the $j$-th row, we replace the tokens in each cell with _NAME_$j$, _ADDR_$j$, _POSTCODE_$j$, _PHONE_$j$. If a table cell is empty, we replace it with an empty token _EMPTY. We do a string match in the transcript and replace the corresponding tokens in transcripts from the table with the special tokens.

Each dialogue on average has 8 turns (16 sentences). We use a vocabulary size of 900, including about 400 table tokens and 500 words.

**Recipes**: We crawl all recipes from `www.allrecipes.com`. There are about $31,000$ recipes in total, and every recipe has a ingredient list and a corresponding recipe. We exclude the recipes that have less than 10 tokens or more than 500 tokens, those recipes take about 0.1% of all data set. On average each recipe has 118 tokens and 9 ingredients. We random shuffle the whole data set and take 80% as training and 10% for validation and test. We use a vocabulary size of 10,000 in the model.

**Coref LM**: We use the Xinhua News data set from Gigaword Fifth Edition and sample 100,000 documents from it that has length in range from 100 to 500. Each document has on average 234 tokens, so there are 23 million tokens in total. We use a tool to annotate all the entity mentions and use the annotation in the training. We take 80% as training and 10% as validation and test respectively. We ignore the entities that have only one mention and for the mentions that have multiple tokens, we take the token that is most frequent in the all the mentions for this entity. After the preprocessing, tokens that are entity mentions take about 10% of all tokens. We use a vocabulary size of 50,000 in the model.

## 4.1 MODEL TRAINING AND EVALUATION

We train all models with simple stochastic gradient descent with clipping. We use a one-layer LSTM for all RNN components. Hyper-parameters are selected using grid search based on the validation set. We use dropout after the input embedding and LSTM output. The learning rate is selected from [0.1, 0.2, 0.5, 1], maximum gradient norm is selected from [1, 2, 5, 10] and drop ratio is selected from [0.2, 0.3, 0.5]. The batch size and LSTM dimension size is slightly different for different tasks so as to make the model fit into memory. The number of epochs to train are different for each task and we drop the learning rate after reaching a given number of epochs. We report the per-word perplexity for all tasks, specifically, we report the perplexity of all words, words that can be generated from reference and non-reference words. For recipe generation, we also generate the recipe using beam size of 10 and evaluate the generated recipe with BLEU.

| model | all | table | table oov | word |
|---|---|---|---|---|
| seq2seq | 1.35±0.01 | 4.98±0.38 | 1.99E7±7.75E6 | 1.23±0.01 |
| table attn | 1.37±0.01 | 5.09±0.64 | 7.91E7±1.39E8 | 1.24±0.01 |
| table pointer | **1.33±0.01** | **3.99±0.36** | **1360 ± 2600** | **1.23±0.01** |
| table latent | 1.36±0.01 | 4.99±0.20 | 3.78E7±6.08E7 | 1.24±0.01 |
| **+ sentence attn** | | | | |
| seq2seq | 1.28±0.01 | 3.31±0.21 | 2.83E9 ± 4.69E9 | 1.19±0.01 |
| table attn | 1.28±0.01 | 3.17±0.21 | 1.67E7±9.5E6 | 1.20±0.01 |
| table pointer | **1.27±0.01** | **2.99±0.19** | **82.86±110** | **1.20±0.01** |
| table latent | 1.28±0.01 | 3.26±0.25 | 1.27E7±1.41E7 | 1.20±0.01 |

Table 4: Dialogue perplexity results. (All means all tokens, table means tokens from table, table oov denotes table tokens that does not appear in the training set, word means non-table tokens). **sentence attn** denotes we use attention mechanism over tokens from past turn. Table pointer and table latent differs in that table pointer, we provide supervised signal on when to generate a table token, while in table latent it is a latent decision.

| model | val | | | | test | | | |
|---|---|---|---|---|---|---|---|---|
| | ppl | | | BLEU | ppl | | | BLEU |
| | all | ing | word | | all | ing | word | |
| seq2seq | 5.60 | 11.26 | **5.00** | 14.07 | 5.52 | 11.26 | **4.91** | 14.39 |
| attn | 5.25 | 6.86 | 5.03 | 14.84 | 5.19 | 6.92 | 4.95 | 15.15 |
| pointer | 5.15 | 5.86 | 5.04 | **15.06** | 5.11 | 6.04 | 4.98 | 15.29 |
| latent | **5.02** | **5.10** | 5.01 | 14.87 | **4.97** | **5.19** | 4.94 | **15.41** |

Table 5: Recipe result, evaluated in perplexity and BLEU score. ing denotes tokens from recipe that appear in ingredients.

| model | val | | | test | | |
|---|---|---|---|---|---|---|
| | all | entity | word | all | entity | word |
| lm | 33.08 | 44.52 | 32.04 | 33.08 | 43.86 | 32.10 |
| pointer | 32.57 | 32.07 | 32.62 | 32.62 | 32.07 | 32.69 |
| pointer + init | **30.43** | **28.56** | **30.63** | **30.42** | **28.56** | **30.66** |

Table 6: Coreference based LM. pointer + init means we initialize the model with the LM weights.

## 4.2 RESULTS AND ANALYSIS

The results for dialogue, recipe generation and coref language model are shown in Table 4, 5 and 6 respectively. We can see from Table 4 that models that condition on table performs better in predicting table tokens in general. Table pointer has the lowest perplexity for token in the table. Since the table token appears rarely in the dialogue, the overall perplexity does not differ much and the non-table tokens perplexity are similar. With attention mechanism over the table, the perplexity of table token improves over basic seq2seq model, but not as good as directly pointing to cells in the table. As expected, using sentence attention improves significantly over models without sentence attention. Surprisingly, table latent performs much worse than table pointer. We also measure the perplexity of table tokens that appear only in test set. For models other than table pointer, because the tokens never appear in training set, the perplexity is quite high, while table pointer can predict these tokens much more accurately. The recipe results in Table 5 in general follows that findings from the dialogue. But the latent model performs better than pointer model since that tokens in ingredients that match with recipe does not necessarily come from the ingredients. Imposing a supervised signal will give wrong information to the model and hence make the result worse. Hence with latent decision, the model learns to when to copy and when to generate it from the vocabulary. The coref LM results are shown in Table 6. We find that coref based LM performs much better on the entities perplexities, but however is a little bit worse than for non-entity words. We found it is an optimization problem and perhaps the model is stuck in local optimum. So we initialize the pointer model with the weights learned from LM, the pointer model performs better than LM both for entity perplexity and non-entity words perplexity.

## 5 RELATED WORK

Recently, there has been great progresses in modeling languages based on neural network, including language modeling (Mikolov et al., 2010; Jozefowicz et al., 2016), machine translation (Sutskever et al., 2014; Bahdanau et al., 2014), question answering (Hermann et al., 2015) etc. Based on the success of seq2seq models, neural networks are applied in modeling chit-chat dialogue (Li et al., 2016; Vinyals & Le, 2015; Sordoni et al., 2015; Serban et al., 2016; Shang et al., 2015) and task oriented dialogue (Wen et al., 2015; Bordes & Weston, 2016; Williams & Zweig, 2016; Wen et al., 2016). Most of the chit-chat neural dialogue models are simply applying the seq2seq models. For the task oriented dialogues, most of them embed the seq2seq model in traditional dialogue systems, in which the table query part is not differentiable. while our model queries the database directly. Recipe generation was proposed in (Kiddon et al., 2016). Their model extents previous work on attention models (Allamanis et al., 2016) to checklists, whereas our work models explicit references to those checklists. Context dependent language models (Mikolov et al., 2010; Ji et al., 2015; Wang & Cho, 2015) are proposed to capture long term dependency of text. There are also lots of works on coreference resolution (Haghighi & Klein, 2010; Wiseman et al., 2016). We are the first to combine coreference with language modeling, to the best of our knowledge. Much effort has been invested in embedding a copying mechanism for neural models (Gülçehre et al., 2016; Gu et al., 2016; Ling et al., 2016). In general, a gating mechanism is employed to combine the softmax over observed words and a pointer network (Vinyals et al., 2015). These gates can be trained either by marginalizing over both outcomes, or using heuristics (e.g. copy low frequency words). Our models are similar to models proposed in (Ahn et al., 2016; Merity et al., 2016), where the generation of each word can be conditioned on a particular entry in knowledge lists and previous words. In our work, we describe a model with broader applications, allowing us to condition, on databases, lists and dynamic lists.

## 6 CONCLUSION

We introduce reference-aware language models which explicitly model the decision of from where to generate the token at each step. Our model can also learns the decision by treating it as a latent variable. We demonstrate on three tasks, table based dialogue modeling, recipe generation and coref based LM, that our model performs better than attention based model, which does not incorporate this decision explicitly. There are several directions to explore further based on our framework. The current evaluation method is based on perplexity and BLEU. In task oriented dialogues, we can also try human evaluation to see if the model can reply users' query accurately. It is also interesting to use reinforcement learning to learn the actions in each step.

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

# A  VISUALIZATION

M:  Hello , welcome to the Cambridge restaurant system ? You can ask for restaurants by area , price range or food type . How may I help you ?
U:  and expensive restaurant that serves tuscan food
M:  Sorry there is no tuscan restaurant in the expensive price range
U:  french food
M:  `cote` is a great restaurant serving french food and it is in the expensive price range
U:  may i have the phone number
M:  The phone number of cote is `01223 311053` .
U:  thank you good bye

(a) Dialogue script

| NAME | PRICE RANGE | FOOD | AREA | ADDRESS | POST CODE | PHONE |
|---|---|---|---|---|---|---|
| charlie chan | cheap | chinese | east | Regent Street City Centre | C.B 2, 1 D.B | 01223 361763 |
| chiquito restaurant bar | expensive | mexican | south | 2G Cambridge Leisure Park Cherry Hinton Road Cherry Hinton | C.B 1, 7 D.Y | 01223 400170 |
| city stop | expensive | food | north | Cambridge City Football Club Milton Road Chesterton | _EMPTY | 01223 363270 |
| clowns cafe | expensive | italian | centre | _EMPTY | C.B 1, 1 L.N | 01223 355711 |
| cocum | expensive | indian | west | 71 Castle Street City Centre | C.B 3, 0 A.H | 01223 366668 |
| cote | expensive | french | centre | Bridge Street City Centre | C.B 2, 1 U.F | 01223 311053 |
| curry garden | expensive | indian | centre | 106 Regent Street City Centre | _EMPTY | 01223 302330 |
| curry king | expensive | indian | centre | 5 Jordans Yard Bridge Street City Centre | C.B 1, 2 B.D | 01223 324351 |
| curry prince | moderate | indian | east | 451 Newmarket Road Fen Ditton | C.B 5, 8 J.J | 01223 566388 |

(b) Attention heat map: `cote` is a great restaurant serving french food and it is in the expensive price range.

| NAME | PRICE RANGE | FOOD | AREA | ADDRESS | POST CODE | PHONE |
|---|---|---|---|---|---|---|
| charlie chan | cheap | chinese | east | Regent Street City Centre | C.B 2, 1 D.B | 01223 361763 |
| chiquito restaurant bar | expensive | mexican | south | 2G Cambridge Leisure Park Cherry Hinton Road Cherry Hinton | C.B 1, 7 D.Y | 01223 400170 |
| city stop | expensive | food | north | Cambridge City Football Club Milton Road Chesterton | _EMPTY | 01223 363270 |
| clowns cafe | expensive | italian | centre | _EMPTY | C.B 1, 1 L.N | 01223 355711 |
| cocum | expensive | indian | west | 71 Castle Street City Centre | C.B 3, 0 A.H | 01223 366668 |
| cote | expensive | french | centre | Bridge Street City Centre | C.B 2, 1 U.F | 01223 311053 |
| curry garden | expensive | indian | centre | 106 Regent Street City Centre | _EMPTY | 01223 302330 |
| curry king | expensive | indian | centre | 5 Jordans Yard Bridge Street City Centre | C.B 1, 2 B.D | 01223 324351 |
| curry prince | moderate | indian | east | 451 Newmarket Road Fen Ditton | C.B 5, 8 J.J | 01223 566388 |

(c) Attention heap map: The phone number of cote is `01223 311053` .

Table 7: Dialogue visualization 1

M: Hello , welcome to the Cambridge restaurant system ? You can ask for restaurants by area , price range or food type . How may I help you ?

U: cheap restaurant

M: What kind of food would you like ?

U: in the west part of town

M: `la margherita` is a nice restaurant in the west of town in the cheap price range

U: address

M: Sure , la margherita is on `15 Magdalene Street City Centre`

U: thank you

M: la margherita is a nice restaurant in the west of town in the cheap price range

U: good bye

(a) Dialogue script

| NAME | PRICE RANGE | FOOD | AREA | ADDRESS | POST CODE | PHONE |
|---|---|---|---|---|---|---|
| india house | expensive | indian | west | 31 Newnham Road Newnham | _EMPTY | 01223 461661 |
| j restaurant | cheap | oriental | centre | 86 Regent Street City Centre | C.B 2, 1 D.P | 01223 307581 |
| jinling noodle bar | moderate | chinese | centre | 11 Peas Hill City Centre | C.B 2, 3 P.P | 01223 566188 |
| kohinoor | cheap | indian | centre | 74 Mill Road City Centre | _EMPTY | 01223 323639 |
| kymmoy | expensive | oriental | centre | 52 Mill Road City Centre | C.B 1, 2 A.S | 01223 311911 |
| la margherita | cheap | italian | west | 15 Magdalene Street City Centre | C.B 3, 0 A.F | 01223 315232 |
| la mimosa | expensive | mediterranean | centre | Thompsons Lane Fen Ditton | C.B 5, 8 A.Q | 01223 362525 |
| la raza | cheap | spanish | centre | 4 - 6 Rose Crescent | C.B 2, 3 L.L | 01223 464550 |
| la tasca | moderate | spanish | centre | 14 -16 Bridge Street | C.B 2, 1 U.F | 01223 464630 |
| lan hong house | moderate | chinese | centre | 12 Norfolk Street City Centre | _EMPTY | 01223 350420 |

(b) Attention heat map: `la margherita` is a nice restaurant in the west of town in the cheap price range

| NAME | PRICE RANGE | FOOD | AREA | ADDRESS | POST CODE | PHONE |
|---|---|---|---|---|---|---|
| india house | expensive | indian | west | 31 Newnham Road Newnham | _EMPTY | 01223 461661 |
| j restaurant | cheap | oriental | centre | 86 Regent Street City Centre | C.B 2, 1 D.P | 01223 307581 |
| jinling noodle bar | moderate | chinese | centre | 11 Peas Hill City Centre | C.B 2, 3 P.P | 01223 566188 |
| kohinoor | cheap | indian | centre | 74 Mill Road City Centre | _EMPTY | 01223 323639 |
| kymmoy | expensive | oriental | centre | 52 Mill Road City Centre | C.B 1, 2 A.S | 01223 311911 |
| la margherita | cheap | italian | west | 15 Magdalene Street City Centre | C.B 3, 0 A.F | 01223 315232 |
| la mimosa | expensive | mediterranean | centre | Thompsons Lane Fen Ditton | C.B 5, 8 A.Q | 01223 362525 |
| la raza | cheap | spanish | centre | 4 - 6 Rose Crescent | C.B 2, 3 L.L | 01223 464550 |
| la tasca | moderate | spanish | centre | 14 -16 Bridge Street | C.B 2, 1 U.F | 01223 464630 |
| lan hong house | moderate | chinese | centre | 12 Norfolk Street City Centre | _EMPTY | 01223 350420 |

(c) Attention heap map: Sure , la margherita is on `15 Magdalene Street City Centre`.

Table 8: Dialogue visualization 2

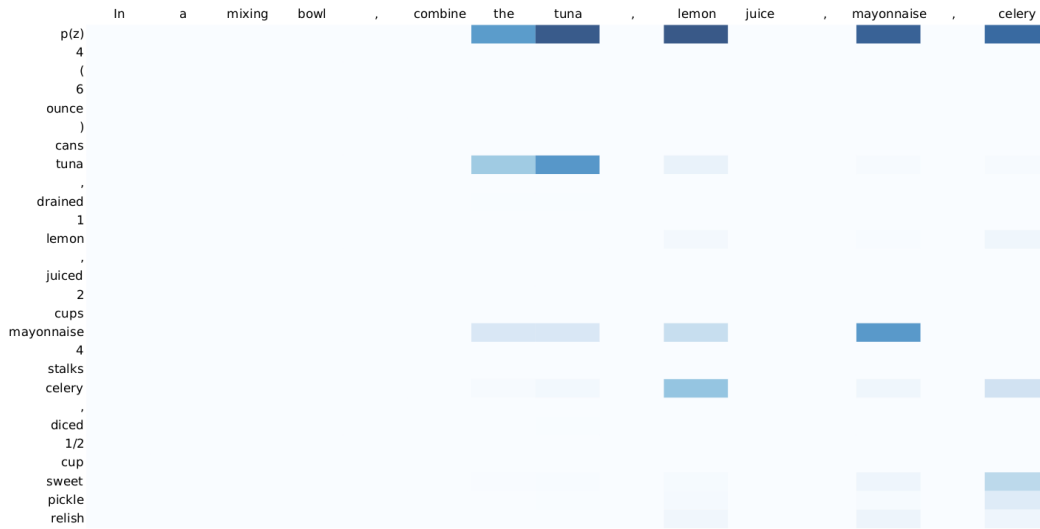

(a) part 1

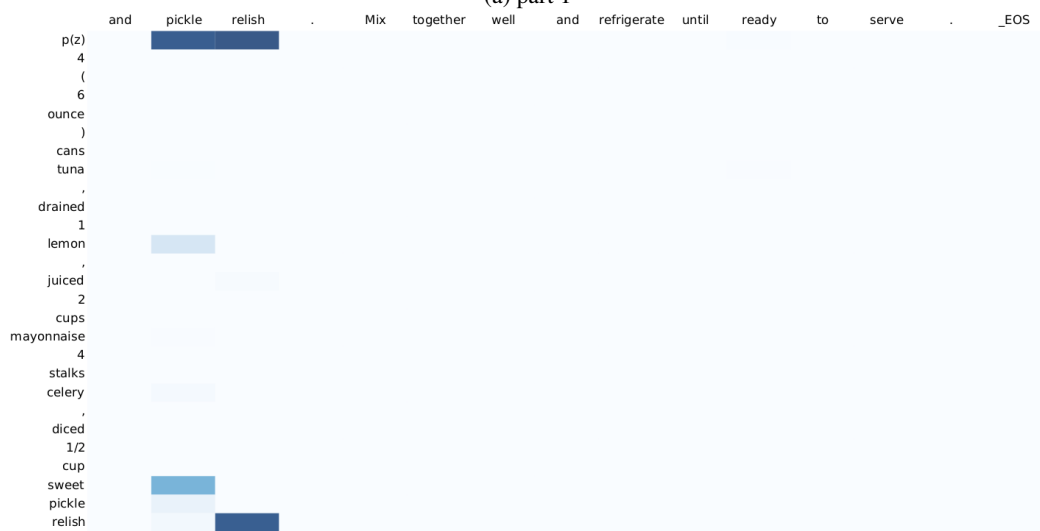

(b) part 2

Figure 6: Recipe heat map example 1. The ingredient tokens appear on the left while the recipe tokens appear on the top. The first row is the $p(z_v|s_v)$.

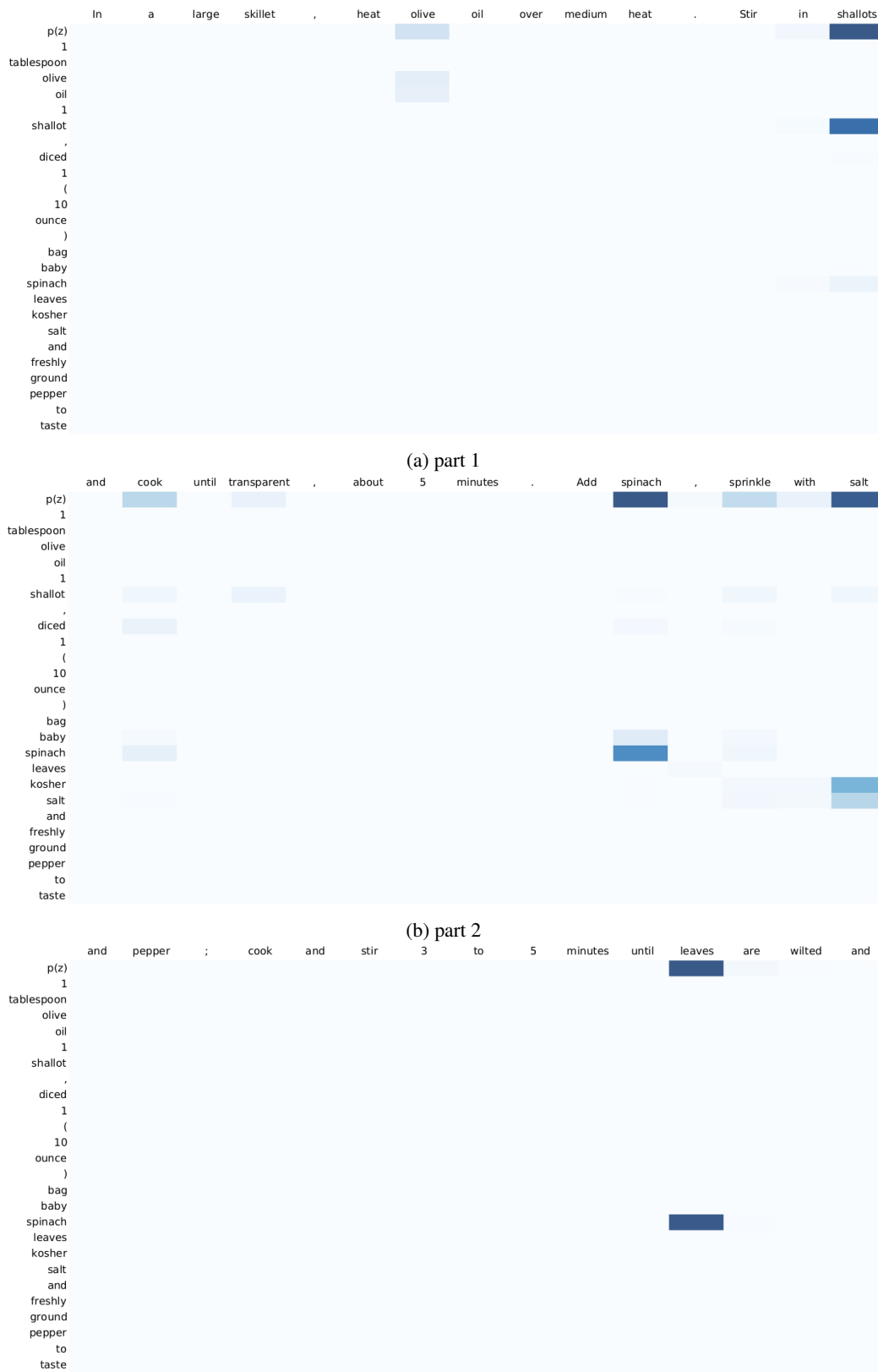

(a) part 1

(b) part 2

(c) part 3

Figure 7: Recipe heat map example 2.

