# Peer review of "Reference-Aware Language Models"

_ICLR 2017 — rejected_

[Public Comment · (anonymous) · 08 Dec 2016]
**Question about the reference-chain**

Hi, can you tell me the reference-chain is obtained by manual annotation or by some automatical tools such stanford corenlp ?

[Official Review · AnonReviewer1 · rating 5 · confidence 4 · 15 Dec 2016]

This paper introduces pointer-network neural networks, which are applied to referring expressions in three small-scale language modeling tasks: dialogue modeling, recipe modeling and news article modeling. When conditioned on the co-reference chain, the proposed models outperform standard sequence-to-sequence models with attention.

The proposed models are essentially variants of pointer networks with copy mechanisms (Gulcehre et al., 2016; Gu et al., 2016; Ling et al., 2016), which have been modified to take into account reference chains. As such, the main architectural novelty lies in 1) restricting the pointer mechanism to focus on co-referenced entities, 2) applying pointer mechanism to 2D arrays (tables), and 3) training with supervised alignments. Although useful in practice, these are minor contributions from an architectural perspective.

The empirical contributions are centred around measuring perplexity on the three language modeling tasks. Measuring perplexity is typical for standard language modeling tasks, but is really an unreliable proxy for dialogue modeling and recipe generation performance. In addition to this, both the dialogue and recipe tasks are tiny compared to standard language modeling tasks. This makes it difficult to evaluate the impact of the dialogue and recipe modeling results. For example, if one was to bootstrap from a larger corpus, it seems likely that a standard sequence-to-sequence model with attention would yield performance comparable to the proposed models (with enough data, the attention mechanism could learn to align referring entities by itself). The language modeling task on news article (Gigaword) seems to yield the most conclusive results. However, the dataset for this task is non-standard and results are provided for only a single baseline. Overall, this limits the conclusions we can draw from the empirical experiments.


Finally, the paper itself contains many errors, including mathematical errors, grammatical errors and typos:
- Eq. (1) is missing a sum over $z_i$.
- "into the a decoder LSTM" -> "into the decoder LSTM"
- "denoted as his" -> "denoted as"
- "Surprising," -> "Surprisingly,"
- "torkens" -> "tokens"
- "if follows that the next token" -> "the next token"
- In the "COREFERENCE BASED LANGUAGE MODEL" sub-section, what does $M$ denote?
- In the sentence: "The attribute of each column is denoted as $s_c, where $c$ is the c-th attribute". For these definitions to be make sense, $s_c$ has to be a one-hot vector. If yes, please clarify this in the text.
- "the weighted sum is performed" -> "the weighted sum is computed"
- "a attribute" -> "an attribute"
- In the paragraph on Pointer Switch, change $p(z_{i,v} |s_{i,v}) = 1$ -> $p(z_{i,v} |s_{i,v}) = 0$.
- In the "Table Pointer" paragraph, I assume you mean outer-product instead of cross-product? Otherwise, I don't see how the equations add up.


Other comments:
- For the "Attention based decoder", is the attention computed using the word embeddings themselves or the hidden states of the sentence encoder? Also, it applied only to the previous turn of the dialogue or to the entire dialogue history? Please clarify this.
- What's the advantage of using an "Entity state update" rule, compared to a pointer network or copy network, which you used in the dialogue and recipe tasks? Please elaborate on this.
- In the Related Work section, the following sentence is not quite accurate: "For the task oriented dialogues, most of them embed the seq2seq model in traditional dialogue systems while our model queries the database directly.". There are task-oriented dialogue models which do query databases during natural language generation. See, for example, "A Network-based End-to-End Trainable Task-oriented Dialogue System" by Wen et al.

[Official Review · AnonReviewer3 · rating 5 · confidence 4 · 17 Dec 2016]
**No Title**

This paper explores 3 language modeling applications with an explicit modeling of reference expressions: dialog, receipt generation and coreferences. While these are important tasks for NLP and the authors have done a number of experiments, the paper is limited for a few reasons:

1. This paper is not clearly written and is pretty hard to follow some details. In particular,  there are many obvious math errors, such as missing the marginalization sum in Eq (1), and P(z_{i,v}...) = 1 (should be 0 here) on page 5, pointer switch section.

2. The major novelty seems to be the 2-dimensional attention from the table and the pointer to the 2-D table. These are more of a customization of existing work to a particular task with 2-D tables as a part of the input to seq2seq model with both attentions and pointer networks.

3. The empirical results are not very conclusive yet, limited by either the relatively small data size, or the lack of well-established baseline for some new applications (e.g., the recipe generation task).

Overall, this paper, as it is for now, is more suitable for a workshop rather than for the main conference.

[Official Review · AnonReviewer2 · rating 6 · confidence 4 · 19 Dec 2016]
**No Title**

This paper presents a new type of language model that treats entity references as latent variables. The paper is structured as three specialized models for three applications: dialog generation with references to database entries, recipe generation with references to ingredients, and text generation with coreference mentions.

Despite some opaqueness in details that I will discuss later, the paper does a great job making the main idea coming through, which I think is quite interesting and definitely worth pursuing further. But it seems the paper was rushed into the deadline, as there are a few major weaknesses.

The first major weakness is that the claimed latent variables are hardly latent in the actual empirical evaluation. As clarified by the authors via pre-review QAs, all mentions were assumed to be given to all model variants, and so, it would seem like an over-claim to call these variables as latent when they are in fact treated as observed variables. Is it because the models with latent variables were too difficult to train right?

A related problem is the use of perplexity as an evaluation measure when comparing reference-aware language models to vanilla language models. Essentially the authors are comparing two language models defined over different event space, which is not a fair comparison. Because mentions were assumed to be given for the reference-aware language models, and because of the fact that mention generators are designed similar to a pointer network, the probability scores over mentions will naturally be higher, compared to the regular language model that needs to consider a much bigger vocabulary set. The effect is analogous to comparing language models with aggressive UNK (and a small vocabulary set) to a language models with no UNK (and a much larger vocabulary set).

To mitigate this problem, the authors need to perform one of the following additional evaluations: either assuming no mention boundaries and marginalizing over all possibilities (treating latent variables as truly latent), or showing other types of evaluation beyond perplexity, for example, BLEU, METEOR, human evaluation etc on the corresponding generation task.

The other major weakness is writing in terms of technical accuracy and completeness. I found many details opaque and confusing even after QAs. I wonder if the main challenge that hinders the quality of writing has something to do with having three very specialized models in one paper, each having a lot of details to be worked out, which may have not been extremely important for the main story of the paper, but nonetheless not negligible in order to understand what is going on with the paper.    Perhaps the authors can restructure the paper so that the most important details are clearly worked out in the main body of the paper, especially in terms of latent variable handling — how to make mention detection and conference resolution truly latent, and if and when entity update helps, which in the current version is not elaborated at all, as it is mentioned only very briefly for the third application (coreference resolution) without any empirical comparisons to motivate the update operation.

[Final Decision · Program Chairs · 06 Feb 2017]
**ICLR committee final decision**

All of the reviewers point out clarity problems; while these may have been resolved in an updated version, the reviewers have not expressed that the matter is resolved. There are several questions raised about the use of perplexity, both whether the comparison is fair, and whether it is a valid proxy for more standard measures in NLP. The former seems to be more of an issue for this area chair, and the discussion did not convince me that it was adequately resolved.